# Aberration of the Green's function estimator in hybridization expansion continuous-time quantum Monte Carlo

Andreas Hausoel[1,2★], Markus Wallerberger[3], Josef Kaufmann[3],
Karsten Held[3] and Giorgio Sangiovanni[2],

**1** Institute for Theoretical Solid State Physics, Leibniz IFW Dresden,
Helmholtzstr. 20, 01069 Dresden, Germany
**2** Institut für Theoretische Physik und Astrophysik
and Würzburg-Dresden Cluster of Excellence ct.qmat,
Universität Würzburg, 97074 Würzburg, Germany
**3** Institute of Solid State Physics, TU Wien, 1040 Vienna, Austria
★ a.hausoel@ifw-dresden.de

November 14, 2022

## Abstract

We describe an aberration of the resampling estimator for the Green's function customarily used in hybridization expansion continuous-time quantum Monte Carlo. It occurs due to Pauli principle constraints in calculations of Anderson impurity models with baths consisting of a discrete energy spectrum. We identify the missing Feynman diagrams, characterize the affected models and discuss implications as well as solutions. This issue does not occur when using worm sampling or in the presence of continuous baths. However certain energy spectra can be inherently close to a discrete limit, and we explain why autocorrelation times can become very large in these cases.

# 1 Introduction

Despite decades of intense research, a generic solution to the quantum many-body problem is still lacking. The use of diagrammatic Monte Carlo techniques, however, has led to significant progress for special cases of interest, such as for the Anderson impurity model (AIM) and its generalizations to correlated molecules and retarded interactions [1].

Diagrammatic Monte Carlo techniques proceed in a three-step fashion: firstly, the action is split into two parts, where one part is solved exactly. Secondly, the other part of the action is treated by expanding an appropriate generating function (such as the partition function or free energy) with respect to it. Thirdly and finally, the resulting probability distribution is sampled using Markov chain Monte Carlo.

Unlike the classical case, in quantum mechanics each observable is operator-valued and comes with its own generating function. Thus after a series expansion, it has its own probability distribution to be sampled. In principle, Monte Carlo algorithms have to sample these distributions separately, e.g., the mean density, the one-particle propagator, and even the propagator evaluated at different times or orbitals.

Two methods are known to deal with this issue: (i) worm sampling [2,3], where one forms the direct sum over the probability spaces of all observables considered, and samples that compound distribution. This quickly leads to an unwieldy number of computations. If the distributions of interest are similar in their structure, which is often the case, with (ii) resampling[1] one can sample only a single distribution and map all other observables to different estimators with respect to that distribution. Resampling is algorithmically simpler but yields an incomplete estimator (and thus wrong results) if the mapping is not surjective. One also runs into autocorrelation problems if the mapping is indeed surjective, but the probability distributions are substantially different.

A widely used, state-of-the-art finite-temperature diagrammatic solver for Anderson impurity models, is continuous-time quantum Monte Carlo in the hybridization expansion [5] (CTHYB), where the partition function is expanded with respect to the bath hybridization. One usually employs resampling for measuring the Green's function, relating each Green's function diagram

---

[1]This is commonly employed for the diagrammatic Monte Carlo calculation of the Green's function in the AIM [4,5].

to a process of "cutting" parts off a diagram in partition function ($Z$) space. This is already known to fail for equal-time correlators, certain higher order Green's functions, and close to the atomic limit. There, worm sampling must be used instead [6]. However, resampling is still widely used because it is believed to succeed away from the aforementioned cases.

In this paper, we identify one more incompleteness of the resampling $G$-estimator for certain finite systems, curtailing the viability of the method in quantum chemistry applications. In analogy to optics we call the phenomenon *aberration*, which means an image being blurred or distorted. Let us note, that the essence of the problem has been already described in one of the authors' thesis [7], and noticed independently of us [8] recently. Here, we further determine all effected systems and show that for certain infinite systems, this form of resampling, while formally consistent, causes the autocorrelation length to grow significantly. Moreover, we point a way out of this problem, by using worm sampling.

The paper is organized as follows: In Section 2 we give a short review of the CTHYB algorithm, before we identify missing Feynman diagrams in systems with finite bath size in Section 3. In Section 4 we show an example and make the link to autocorrelation times. Finally in Section 5, we show a system with infinite bath size and autocorrelation problems, before we conclude in Section 6.

## 2   CTHYB and the measurement of the Green's function

In this section we repeat the basic concepts of CTHYB, namely the expansion formulas and how they relate to Feynman diagrams, as well as the differences between the two types of measuring the Green's function: $Z$-sampling, corresponding to (ii) resampling, and $G$-sampling, corresponding to (i) worm sampling. For more details we refer the reader to the corresponding literature[2].

### 2.1   The expansion formulas and Feynman diagrams

The Hamiltonian of the multi-orbital AIM reads

$$
\begin{aligned}
\hat{H}_{\mathrm{AIM}} &= \hat{H}_{\mathrm{bath}} + \hat{H}_{\mathrm{hyb}}^{\dagger} + \hat{H}_{\mathrm{hyb}} + \hat{H}_{\mathrm{loc}} \\
&= \sum_{p\mu} \epsilon_{p\mu} \hat{a}_{p\mu}^{\dagger} \hat{a}_{p\mu} + \sum_{p\mu\nu} V_{p\mu\nu}^{*} \hat{a}_{p\mu}^{\dagger} \hat{c}_{\nu} + \sum_{p\mu\nu} V_{p\mu\nu} \hat{c}_{\nu}^{\dagger} \hat{a}_{p\mu} \\
&\quad + \hat{H}_{\mathrm{loc}}[\hat{c}^{\dagger}, \hat{c}].
\end{aligned}
\tag{1}
$$

Operators $\hat{c}_{\nu}^{\dagger}$ ($\hat{c}_{\nu}$) create (annihilate) electrons on the impurity with flavor $\nu$, whereas operators $\hat{a}_{p\mu}^{\dagger}$ ($\hat{a}_{p\mu}$) create (annihilate) electrons on the $p$-th bath site, which has an energy of $\epsilon_{p\mu}$ and belongs to impurity flavor $\mu$. In the second and third term of Eq. (1), each impurity flavor $\nu$ couples to its own non-interacting bath sites with amplitudes $V_{p\nu\nu}$ (diagonal hybridization), but may also couple to the bath sites of other impurity flavors $\mu$ via $V_{p\mu\nu}$ with $\mu \neq \nu$ (off-diagonal hybridization). The fourth term $\hat{H}_{\mathrm{loc}}[\hat{c}^{\dagger}, \hat{c}] = -t_{\mu\nu} \hat{c}_{\mu}^{\dagger} \hat{c}_{\nu} + U_{\kappa\lambda\mu\nu} \hat{c}_{\kappa}^{\dagger} \hat{c}_{\lambda}^{\dagger} \hat{c}_{\nu} \hat{c}_{\mu}$ contains the one- and two-particle interaction on the impurity.

---

[2]Our notation is based on chapter II of Reference 6 and chapter II of Reference 9. For all details take a look at Reference 1.

The CTHYB expansion of the partition function is [1,9]

$$Z = \sum_{k=0}^{\infty} \frac{1}{k!} \int \mathrm{d}^k \mathcal{C} \int \mathrm{d}^k \mathcal{C}' \, w_{\mathrm{loc}}(\mathcal{C}, \mathcal{C}') w_{\mathrm{bath}}(\mathcal{C}, \mathcal{C}'), \tag{2}$$

i.e., an integral over configurations $(\mathcal{C}, \mathcal{C}')$ of an appropriate weight function. More specifically, one defines a configuration $\mathcal{C} = \{(\nu_1, \tau_1), \ldots, (\nu_k, \tau_k)\}$ as a set of times and flavors for the creation operators, and $\mathcal{C}' = \{(\nu'_1, \tau'_1), \ldots, (\nu'_k, \tau'_k)\}$ as the corresponding set for the annihilation operators, where $k$ is the expansion order. The integral in Eq. (2) thus is

$$\int \mathrm{d}^k \mathcal{C} \equiv \sum_{\nu_1} \cdots \sum_{\nu_k} \int_0^{\beta} \mathrm{d}\tau_1 \int_0^{\beta} \mathrm{d}\tau_2 \cdots \int_0^{\beta} \mathrm{d}\tau_k \tag{3}$$

and the local weight is given by

$$w_{\mathrm{loc}}(\mathcal{C}, \mathcal{C}') = \mathrm{Tr}_c \left[ e^{-\beta \hat{H}_{\mathrm{loc}}} T_\tau \prod_{i=1}^{k} \hat{c}^{\dagger}_{\nu_i}(\tau_i) \hat{c}_{\nu'_i}(\tau'_i) \right]. \tag{4}$$

Here, the trace $\mathrm{Tr}_c[\ldots] = \sum_s \langle s | \ldots | s \rangle$ is computed over a complete many-body basis of the impurity. For two flavors (one spin-1/2 band) such a basis would be $|s\rangle \in \{|0\rangle, |\uparrow\rangle, |\downarrow\rangle, |\uparrow\downarrow\rangle\}$. The argument of the trace is a time-ordered product of impurity operators, whose time-evolution is governed by $\hat{H}_{\mathrm{loc}}$ via $\hat{c}_\nu(\tau) = e^{\hat{H}_{\mathrm{loc}}\tau} \hat{c}_\nu e^{-\hat{H}_{\mathrm{loc}}\tau}$.

The bath weight describes the retardation effect of the bath on the impurity and is given by

$$w_{\mathrm{bath}}(\mathcal{C}, \mathcal{C}') = \det \begin{pmatrix} \Delta_{\nu_1 \nu'_1}(\tau_1 - \tau'_1) & \cdots & \Delta_{\nu_1, \nu'_k}(\tau_1 - \tau'_k) \\ \vdots & \ddots & \vdots \\ \Delta_{\nu_k \nu'_1}(\tau_k - \tau'_1) & \cdots & \Delta_{\nu_k, \nu'_k}(\tau_k - \tau'_k) \end{pmatrix}, \tag{5}$$

where the matrix elements are $\Delta_{ij} = \Delta_{\nu_i, \nu'_j}(\tau_i - \tau'_j)$ with $(\nu_i, \tau_i) \in \mathcal{C}$ and $(\nu'_j, \tau'_j) \in \mathcal{C}'$. The propagator is the hybridization function [1]

$$\Delta_{\nu \nu'}(\tau) = \sum_{p=1}^{N_p} \frac{V_{p\mu\nu} V^*_{p\mu\nu'}}{e^{\beta \epsilon_{p\mu}} + 1} \times \begin{cases} e^{\epsilon_{p\mu}\tau}, & \text{if } \tau > 0, \\ -e^{\epsilon_{p\mu}(\beta - \tau)}, & \text{if } \tau < 0, \end{cases} \tag{6}$$

which is a sum of the non-interacting Green's functions of the $N_p$ bath sites, weighted with the hopping amplitudes $V_{p\mu\nu} V^*_{p\mu\nu'}$ ($N_p$ can in principle be infinite). Therefore it contains three processes combined: the hopping from impurity to bath, propagation through the bath, and hopping back from bath to impurity.

Let us take the partition function configuration $\hat{c}^{\dagger}_{\uparrow}(\tau_1) \hat{c}^{\dagger}_{\downarrow}(\tau_2) \hat{c}_{\downarrow}(\tau'_1) \hat{c}_{\uparrow}(\tau'_2)$ of an AIM with two flavors $\uparrow$ and $\downarrow$ as example of a $w_{\mathrm{bath}}$, see Fig. 1 (a). This corresponds to $\mathcal{C} = \{(\downarrow, \tau_1), (\uparrow, \tau_2)\}$ and $\mathcal{C}' = \{(\downarrow, \tau'_1), (\uparrow, \tau'_2)\}$ in the abbreviated notation. The filled (empty) diamonds indicate the annihilation (creation) of an electron on the impurity. The dashed lines attached to an operator represent a connection of this operator to the bath. In our example a pair of an impurity electron and bath hole is created at $\tau_2$. The bath hole propagates to $\tau'_2$, where it annihilates with another impurity electron; the propagator for these three processes combined is $\Delta_{\uparrow\uparrow}(\tau_2 - \tau'_2)$. The same happens with another hole from $\tau_1$ to $\tau'_1$ via $\Delta_{\downarrow\downarrow}(\tau_1 - \tau'_1)$. A second possibility is that the hole

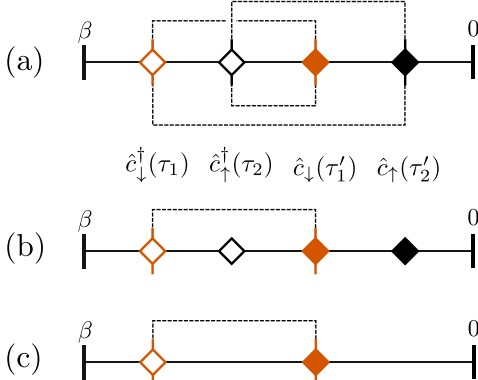

Figure 1: (a), (c) $Z$-configurations for an AIM with one orbital. Empty diamonds are creation operators, filled diamonds annihilation operators, and color denotes impurity flavor. The dashed lines mean a propagation of the electron through the bath. (b) Derived Green's function configuration obtained either (i) from (c) through worm sampling inserting two (black) worm operators at $\tau_2$ and $\tau_2'$, or (ii) from (a) through $Z$-sampling removing the hybridization lines from the black operators.

at $\tau_2'$ propagates to $\tau_1'$ via $\Delta_{\uparrow\downarrow}(\tau_2 - \tau_1')$, and the other hole from $\tau_1$ to $\tau_2'$ via $\Delta_{\downarrow\uparrow}(\tau_1 - \tau_2')$. All these propagations can compactly be written as a determinant

$$w_{\text{bath}}(\mathcal{C}, \mathcal{C}') = \det \begin{pmatrix} \Delta_{\downarrow\downarrow}(\tau_1 - \tau_1') & \Delta_{\downarrow\uparrow}(\tau_1 - \tau_2') \\ \Delta_{\uparrow\downarrow}(\tau_2 - \tau_1') & \Delta_{\uparrow\uparrow}(\tau_2 - \tau_2') \end{pmatrix}, \tag{7}$$

which corresponds to the application of Wick's theorem to the non-interacting bath.

## 2.2 Resampling and worm sampling

Now we discuss the two types of measuring the Green's function. The hybridization expansion of the Green's function is

$$G_{\nu\nu'}(\tau - \tau') = \frac{1}{Z} \sum_{\mathcal{C}} w_{\text{loc}}(\mathcal{C} \cup \{(\nu, \tau)\}, \mathcal{C}' \cup \{(\nu', \tau')\})$$
$$\times w_{\text{bath}}(\mathcal{C}, \mathcal{C}'). \tag{8}$$

(i) Worm sampling takes a partition function configuration $(\mathcal{C}, \mathcal{C}')$, which was generated by the Markov chain, and adds $\{(\nu, \tau)\}$ to the set of creators, and $\{(\nu', \tau')\}$ to the set of annihilators in the local weight only (indicated by the union symbol). This is depicted in Fig. 1 (b), with Green's functions operators $\hat{c}_\uparrow^\dagger(\tau_2)$ and $\hat{c}_\uparrow(\tau_2')$ added to the configuration (c). Sampling this expansion is referred to as worm sampling or $G$-sampling [6].

(ii) The standard way to measure the Green's function ($Z$-sampling) is instead a form of resampling. The Markov chain produces partition function configurations with weight $w_{\text{loc}}(\mathcal{C}, \mathcal{C}') w_{\text{bath}}(\mathcal{C}, \mathcal{C}')$,

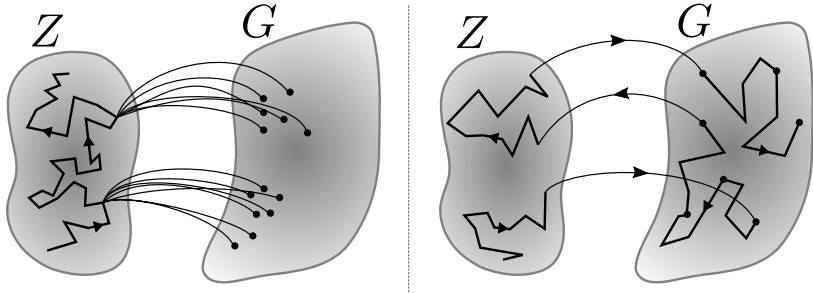

Figure 2: Illustration of the two sampling procedures to measure the Green's function. Arrows indicate the direction that the random walker moves, black circles give the points in $G$-space where the Green's function is measured. (ii) In $Z$-sampling (left) the sampling only occurs in $Z$-space, and for generating Green's function configurations the $Z$-configuration is changed (two hybridization lines are removed). (i) In $G$-sampling (right) also $G$-space is sampled.

from which Green's function configurations are created via

$$
\begin{aligned}
G_{\alpha\alpha'}(\tau) = \frac{1}{Z} \sum_{\mathcal{C}} & w_{\mathrm{loc}}(\mathcal{C}, \mathcal{C}') w_{\mathrm{bath}}(\mathcal{C}, \mathcal{C}') \\
& \times \sum_{n,m=1}^{k} \frac{w_{\mathrm{bath}}(\mathcal{C} \setminus \{(\alpha_n, \tau_n)\}, \mathcal{C}' \setminus \{(\alpha'_m, \tau'_m)\})}{w_{\mathrm{bath}}(\mathcal{C}, \mathcal{C}')} \\
& \times \delta^{-}\big[\tau - (\tau_n - \tau'_m)\big] \delta_{\alpha\alpha_n} \delta_{\alpha'\alpha'_m}.
\end{aligned}
\tag{9}
$$

The object after the sum $\sum_{nm}$ is the estimated quantity, where the original bath weight $w_{\mathrm{bath}}(\mathcal{C}, \mathcal{C}')$ is replaced by a bath weight, where $\{(\alpha_n, \tau_n)\}$ has been removed from the creator vertices and $\{(\alpha'_m, \tau'_m)\}$ from the annihilator vertices. We indicate this by the set difference. The antiperiodic Dirac comb is defined by $\delta[\tau] = \sum_{n\in\mathbb{Z}} (-1)^n \delta(\tau - n\beta)$. This removal leaves behind two operators in the local weight without hybridization lines, making them the Green's function operators. To exploit the full information of a $Z$-configuration, the procedure of removing the hybridization lines is applied to all possible pairs of annihilation and creation operators by the sum $\sum_{nm}$. The Green's function configuration in Fig. 1 (b) was constructed by removing the hybridization lines of two operators $\hat{c}_\uparrow^\dagger(\tau_2)$ and $\hat{c}_\uparrow(\tau'_2)$ compared to (a). This also means, that to each $G$-configuration uniquely belongs a $Z$-configuration.

Finally, let us illustrate the two sampling procedures in Fig. 2. In $Z$-sampling the sampling solely occurs in $Z$-space; for the measurement, Green's function configurations are created from $Z$-configurations. In $G$-sampling the random walk moves between $Z$-space and $G$-space and samples and measures in both.

## 3 Incompleteness of the resampling $G$-estimator

Here we work out the detected incompleteness of the resampling estimator for the Green's function. Equation (9) looks already dangerous: suppose $w_{\mathrm{bath}}(\mathcal{C}, \mathcal{C}')$ is zero, then the $Z$-configuration $w_{\mathrm{loc}}(\mathcal{C}, \mathcal{C}') w_{\mathrm{bath}}(\mathcal{C}, \mathcal{C}')$ has zero weight and will never be reached. Second, if removing the hy-

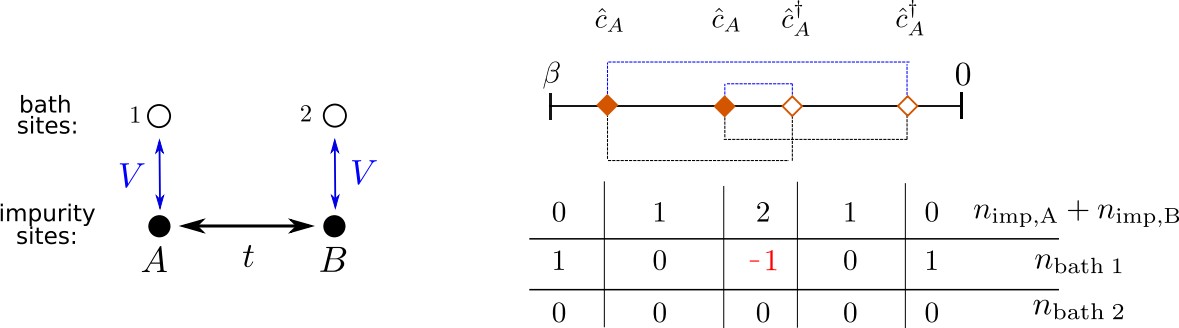

Figure 3: Left: sketch of an impurity with two flavors $A$ and $B$ (black circles), which are connected via a hopping $t$. Each of them has a single bath site (empty circle), to which it couples with amplitude $V$.

Top right: a critical $Z$-configuration of impurity operators (the dashed lines indicate operators connected to the bath).

Bottom right: a table, which shows the total impurity occupation $n_{\mathrm{imp,A}} + n_{\mathrm{imp,B}}$ and bath occupations $n_{\mathrm{bath\ 1}}$ and $n_{\mathrm{bath\ 2}}$ for the $Z$-configuration depicted in the middle as a function of imaginary time.

bridization lines of two operators gives a nonzero weight

$$w_{\mathrm{bath}}(\mathcal{C}\backslash\{(\alpha_n, \tau_n)\}, \mathcal{C}'\backslash\{(\alpha'_m, \tau'_m)\}) \tag{10}$$

for a specific pair of $n$ and $m$, these Green's function configurations will not be generated by $Z$-sampling. It is known that this is the case in the atomic limit [6], as well as for correlators with equal-time operators, which never occur when sampling the times continuously.

Here we show that it also happens in systems with a finite number of non-degenerate bath sites, where the bath can only host a finite number of electrons due to Pauli's exclusion principle. Suppose a $Z$-configuration deposits one electron more in the bath than the bath can hold. Then the weight of that $Z$-configuration becomes exactly zero, but derived $G$-configurations, which depose one electron less in the bath and thus can have non-zero weight, are "missed". In the following we call these $Z$-diagrams "*critical*", since they would be necessary for obtaining all proper $G$-configurations. The AIMs and diagrams affected by this issue can be exactly characterized, which we will do in the following. Two ideas of curing the problem are discussed, namely adding offdiagonal hybridizations and adding more bath sites, before we formulate the general criterion to recognize models, for which the resampling $G$-estimator is incomplete.

## 3.1 The simplest model: one bath site per impurity site

In Figure 3 we investigate the arguably smallest system showing the incompleteness: an impurity *cluster* of size $N = 2$ (two flavors $A$ and $B$, drawn by black circles), connected by a single-particle hopping $t$. Impurity flavor $A$ has a single bath site with label 1, and flavor $B$ one with label 2 (drawn by empty circles), to which they couple with amplitude $V$.

Figure 3 also shows an example $Z$-configuration, thus all the impurity operators are connected to the bath. We analyze the occupations of impurity and bath sites. Reading from right to left, two electrons of flavor $A$ are created, which raises the impurity occupation $n_{\mathrm{imp}} = n_{\mathrm{imp,A}} + n_{\mathrm{imp,B}}$ from zero to two. This is possible, since the first operator creates the single occupied state $|A\rangle = \hat{c}^\dagger_A |0\rangle$,

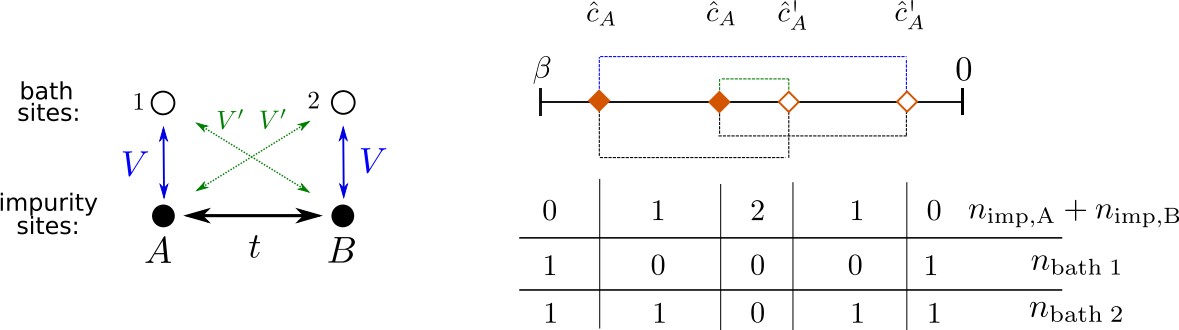

Figure 4: Same as Fig. 3 but now with an additional off-diagonal hybridizations $V'$. In this case, the Pauli violation of Fig. 3 is avoided.

which the time evolution delocalizes via $e^{\hat{H}\tau}|A\rangle = \alpha|A\rangle + \alpha'|B\rangle$, with $\alpha, \alpha' \in \mathbb{C}$, due to the single particle hopping $t$ (even though the occupation number basis is not favorable for implementation, we use this basis for the discussion here). Therefore the application of the second operator can give a doubly occupied impurity state $|AB\rangle = \hat{c}_A^\dagger |A\rangle + \hat{c}_A^\dagger |B\rangle$ with nonzero amplitude. The same way two electrons of flavor $A$ can be annihilated, decreasing the impurity occupation from 2 to 0.

However, there is now no possible way for bath site 1 to receive the two electrons of flavor $A$. One either had to end at $\tau = \beta$ with two electrons in bath 1, which would violate Pauli's principle; or allow an occupation of -1 as indicated in red in Fig. 3, which is not possible either. This makes the weight of this $Z$ configuration zero. In App. C we discuss in more detail, how Pauli's principle is implemented in the language of effective propagators.

If we now create a $G$-configuration out of the $Z$-diagram in Fig. 3, we obtain a valid non-zero $G$-diagram with only one $\hat{c}_A^\dagger$ and $\hat{c}_A$ connected to the bath. Since it has to be accessed via the zero-weight $Z$-diagram, it is missed in $Z$-sampling for this discrete model. Let us stress that this $G$-diagram cannot be created out of another $Z$-diagram, since in $Z$-sampling every $G$-diagram uniquely belongs to a $Z$-diagram.

## 3.2 Adding offdiagonal hybridizations

Adding off-diagonal hybridization removes the problem, since the second electron of impurity flavor $A$ can be delivered by bath site number 2 (see Fig. 4). Therefore, critical diagrams do not exist in clusters with off-diagonal hybridizations, that have more or the same number of bath sites compared to the cluster.

## 3.3 Adding more bath sites

One might think adding more bath sites to the system in Fig. 3 resolves the problem of missing $G$-diagrams; this however is only partially true, as we show in this section.

In Fig. 5 we see, that by adding an additional bath site per impurity flavor, the diagram from the previous Section 3.1 is not critical any more. For reasons that will become clear immediately, we also added to the trace two operators of flavor $B$.

However, for this system a critical diagram can still be constructed by adding a $\hat{c}_A \hat{c}_B^\dagger$ on the left, and a $\hat{c}_B \hat{c}_A^\dagger$ on the right (gray boxes in Fig. 6). This leads to a construction principle for critical $Z$-diagrams for this system with $N_{\text{baths}}$ non-degenerate bath sites per impurity flavor: start with $\hat{c}_A \hat{c}_A^\dagger$ and add $N_{\text{baths}}$ times a block of $\hat{c}_A \hat{c}_B^\dagger$ on the left, and $N_{\text{baths}}$ times a block of $\hat{c}_B \hat{c}_A^\dagger$ on the right.

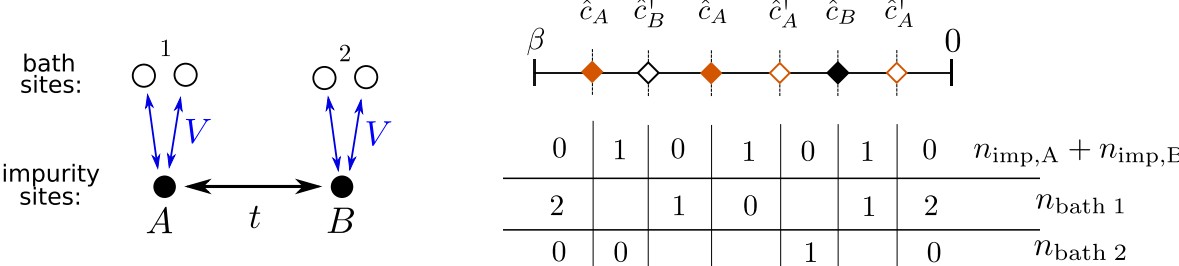

Figure 5: A non-critical $Z$-diagram for a system with 2 bath sites per impurity flavor. For clarity the dashed lines symbolizing propagations through the bath have been omitted here.

For an impurity cluster, where one impurity site has $N_{\text{baths}}$ non-degenerate bath sites, a $Z$-diagram needs $N_{\text{baths}} + 1$ impurity creators (annihilators) in a row to be critical, without annihilators (creators) of the same flavor in between. However, if $N_{\text{baths}} + 1$ is larger than the cluster size, the local trace is not necessarily zero for exceeding the maximal (subceeding zero) occupation, since this can get compensated by impurity operators of the other flavor.

Combinatorics suggest that for systems with a large number of non-degenerate bath sites, critical diagrams are much more rare compared to systems with only a few non-degenerate bath sites; we will show this explicitly in Section 4.

### 3.4 General criterion for incomplete resampling $G$-estimator

In summary, the following conditions suffice for critical diagrams to exist: The impurity has a cluster (impurity sites connected by one- or two-particle interaction terms) of size greater or equal to 2; one of the cluster sites has a finite bath, and there is no hybridization of this finite bath to other flavors.

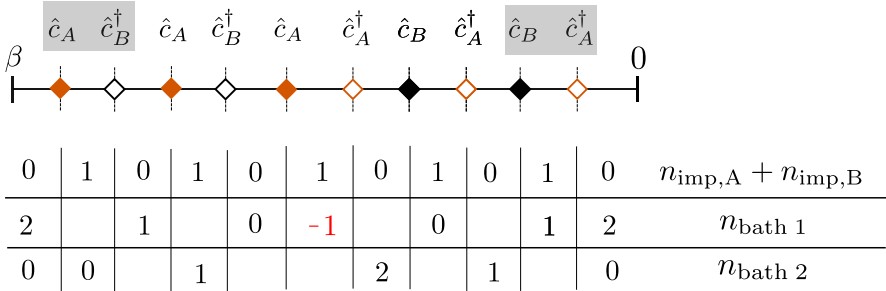

Figure 6: A critical diagram for a system with 2 non-degenerate bath sites per impurity flavor.

## 4 Numerical analysis

In this section we provide numerical evidence for the considerations formulated in the Section 3. For the codes employed, see Appendix A. First we show that $Z$-sampling gives wrong Green's

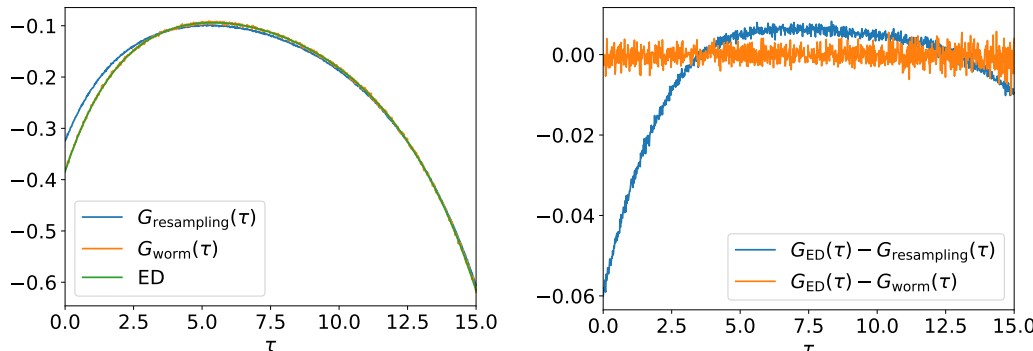

Figure 7: Left: Green's function $G_{\uparrow\uparrow}(\tau)$ for model Eq. (11) with (i) worm sampling and (ii) resampling compared to ED. Right: Differences of the QMC results to ED.

functions for the system in Fig. 3. Then we demonstrate that critical diagrams are much less important in systems with more non-degenerate bath sites. Finally we interpret this outcome by analyzing the autocorrelation times of critical diagrams.

## 4.1 The simplest model

The Hamiltonian of the system under consideration reads

$$
\begin{aligned}
\hat{H} &= \begin{pmatrix} \hat{c}_{\uparrow}^{\dagger} & \hat{c}_{\downarrow}^{\dagger} & \hat{a}_{\uparrow}^{\dagger} & \hat{a}_{\downarrow}^{\dagger} \end{pmatrix}
\begin{pmatrix}
E_{\uparrow} & t & V & 0 \\
t & E_{\downarrow} & 0 & V \\
V & 0 & \epsilon_{\uparrow} & 0 \\
0 & V & 0 & \epsilon_{\downarrow}
\end{pmatrix}
\begin{pmatrix} \hat{c}_{\uparrow} \\ \hat{c}_{\downarrow} \\ \hat{a}_{\uparrow} \\ \hat{a}_{\downarrow} \end{pmatrix} \\
&= \sum_{\sigma} \left[ E_{\sigma}\hat{n}_{\sigma} + t\hat{c}_{\sigma}^{\dagger}\hat{c}_{\bar{\sigma}} + \epsilon_{\sigma}\hat{a}_{\sigma}^{\dagger}a_{\sigma} + V(\hat{c}_{\sigma}^{\dagger}\hat{a}_{\sigma} + \hat{a}_{\sigma}^{\dagger}\hat{c}_{\sigma}) \right].
\end{aligned}
\tag{11}
$$

The two impurity flavors $E_{\sigma}$ could differ by any quantum number like orbital, spin, or a combination thereof. In order to deal with one-orbital models, we choose this quantum number to be the spin. The aberration of the $Z$-estimator does not require electron-electron interaction, hence for simplicity we consider the system to be non-interacting. Let us note, however, that interactions beyond density-density terms can substitute the role of $t$, as they allow as well moving electrons from one impurity flavor to another, and thus to add consecutively a second electron with the same impurity flavor.

The general criterion formulated at the end of Sec. 3 is clearly violated here. We have an impurity cluster of size two, connected to two single bath sites.

Fig. 7 confirms this numerically for values of $E_{\uparrow} = E_{\downarrow} = 0.0$, $\epsilon_{\uparrow} = 0.2$ $\epsilon_{\downarrow} = -0.2$, $t = 0.2$, $V = 0.2$ and an inverse temperature of $\beta = 15.0$. The Green's functions of $Z$-sampling are clearly wrong compared to $G$-sampling or exact diagonalization (ED). Even the property

$$
G_{\sigma\sigma'}(\tau = 0^{+}) + G_{\sigma\sigma'}(\tau = \beta^{-}) = \delta_{\sigma\sigma'}
\tag{12}
$$

from the anticommutation relations of Fermionic operators is violated in $Z$-sampling, as one can see from deviation of the results from resampling compared to ED in Fig. 7.

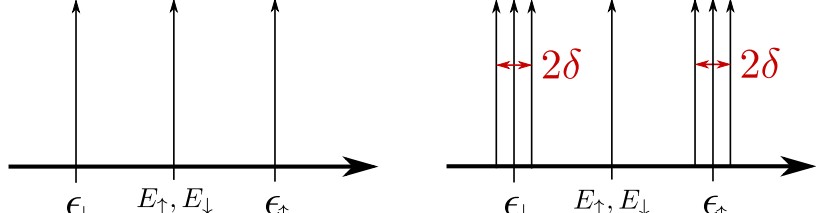

Figure 8: Depiction of the discrete energy levels of Hamiltonian (14) for zero and finite $\delta$.

## 4.2 Autocorrelation time analysis

We know from Section 4.1, that resampling for model Eq. (11) misses all critical diagrams, which have two annihilation (creation) operators of the same flavor consecutively in a row (operators of the other flavor may be within that row), or equivalently, whose bath occupation exceeds (subceeds) 1(0). Let's decompose the Green's function into two parts

$$G(\tau) = G_{\text{crit}}(\tau) + G_{\text{noncrit}}(\tau), \tag{13}$$

where $G_{\text{crit}}(\tau)$ contains all $G$-diagrams derived from the critical $Z$-diagrams for model Eq. (11). This way the influence of the critical diagrams and their autocorrelation times can be analyzed.

For this purpose, we continuously interpolate between a system with a single bath site per impurity flavor, and one with three non-degenerate bath sites per impurity flavor. This can be achieved by splitting the single bath site into three and shifting their energy levels apart by a parameter $\delta$ (cf. Fig. 8):

$$
\hat{H}'[\delta] =
\begin{pmatrix}
\hat{c}^\dagger_\uparrow \\
\hat{c}^\dagger_\downarrow \\
\hat{a}^\dagger_{1\uparrow} \\
\hat{a}^\dagger_{2\uparrow} \\
\hat{a}^\dagger_{3\uparrow} \\
\hat{a}^\dagger_{1\downarrow} \\
\hat{a}^\dagger_{2\downarrow} \\
\hat{a}^\dagger_{3\downarrow}
\end{pmatrix}^{\mathrm{T}}
\cdot
\begin{pmatrix}
E_\uparrow & t & v & v & v & 0 & 0 & 0 \\
t & E_\downarrow & 0 & 0 & 0 & v & v & v \\
v & 0 & \epsilon_\uparrow - \delta & 0 & 0 & 0 & 0 & 0 \\
v & 0 & 0 & \epsilon_\uparrow & 0 & 0 & 0 & 0 \\
v & 0 & 0 & 0 & \epsilon_\uparrow + \delta & 0 & 0 & 0 \\
0 & v & 0 & 0 & 0 & \epsilon_\downarrow - \delta & 0 & 0 \\
0 & v & 0 & 0 & 0 & 0 & \epsilon_\downarrow & 0 \\
0 & v & 0 & 0 & 0 & 0 & 0 & \epsilon_\downarrow + \delta
\end{pmatrix}
\cdot
\begin{pmatrix}
\hat{c}_\uparrow \\
\hat{c}_\downarrow \\
\hat{a}_{1\uparrow} \\
\hat{a}_{2\uparrow} \\
\hat{a}_{3\uparrow} \\
\hat{a}_{1\downarrow} \\
\hat{a}_{2\downarrow} \\
\hat{a}_{3\downarrow}
\end{pmatrix}.
\tag{14}
$$

We further discuss this construction in appendix D. For consistency with the previous model Eq. (11) the new hybridization is $v = V/\sqrt{3}$. One can verify by using the resolvent expression $G = (\mathbb{1}i\omega - \hat{H})^{-1}$ or calculating the hybridization function, that the systems $\hat{H}$ and $\hat{H}'[\delta = 0]$ are identical.

A scan over the parameter $\delta$ is shown in Fig. 9 for $\delta \in \{0.05, 0.01, 0.003, 0.001, 0.0\}$. Since the number of Monte Carlo steps and measurements is the same for all panels and the systems are very similar, the difference in the noise can be regarded as an autocorrelation effect. $Z$-sampling agrees with ED for the system with $\delta = 0.05$; this system can be considered as having three non-degenerate bath sites per impurity flavor, therefore the contributions of critical diagrams for $N_{\text{baths per flavor}} = 1$ are all produced frequently in $Z$-sampling. The critical diagrams for $N_{\text{baths per flavor}} = 3$ are not produced, but their absence is clearly not visible with the Monte Carlo precision of this calculation. Upon decreasing $\delta$ the noise increases strongly, because $Z$-sampling

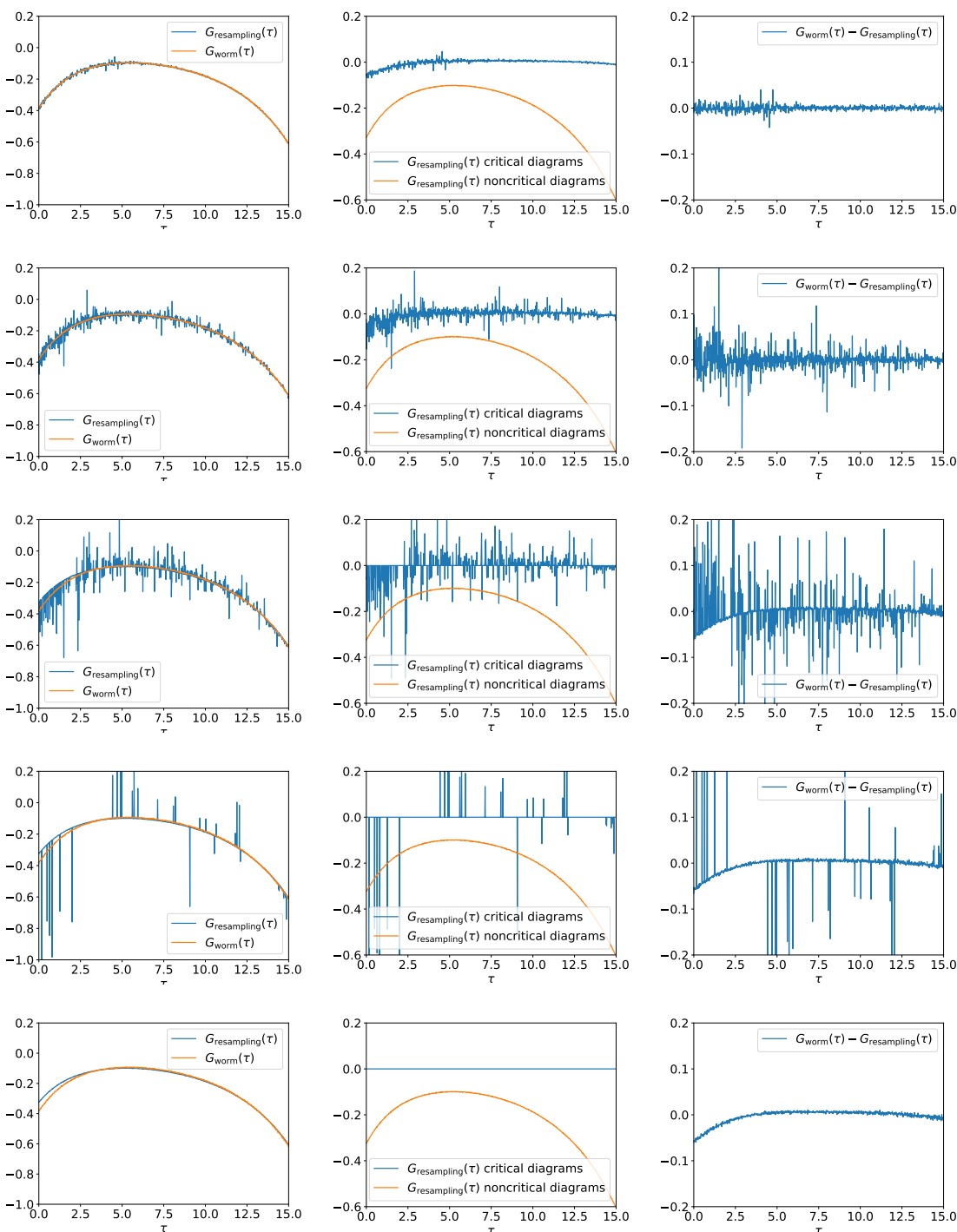

Figure 9: From top to bottom: Hamiltonian $H'[\delta]$ of Eq. (14) for $\delta = 0.05, 0.01, 0.003, 0.001, 0.0$. The left column shows the Green's functions for resampling and worm sampling, the middle column a distribution of the resampling Green's function in critical and noncritical diagrams, and the right column shows the difference between worm and resampling results.

is less and less able to produce the critical diagrams for $N_{\text{baths per flavor}} = 1$. For $\delta = 0.00$ none of those can be produced by the $Z$-sampling any more. Their contribution is zero without noise and the result is wrong. In other words, upon approaching the $N_{\text{baths per flavor}} = 1$ limit, the auto-correlation times of the critical diagrams increases, and becomes infinite when reaching the limit. The curve for the noncritical diagrams is smooth all the time, which means their autocorrelation times are small and independent of $\delta$. Worm sampling is instead able to produce all $G$-diagrams without problems.

The same observation can be made if we –instead of $\delta$– extend the Hamiltonian of Eq. (11) by an offdiagonal hybridization $V'$:

$$\hat{H}''[V'] = \begin{pmatrix} \hat{c}_\uparrow^\dagger & \hat{c}_\downarrow^\dagger & \hat{a}_\uparrow^\dagger & \hat{a}_\downarrow^\dagger \end{pmatrix} \begin{pmatrix} E_\uparrow & t & V & V' \\ t & E_\downarrow & V' & V \\ V & V' & \epsilon_\uparrow & 0 \\ V' & V & 0 & \epsilon_\downarrow \end{pmatrix} \begin{pmatrix} \hat{c}_\uparrow \\ \hat{c}_\downarrow \\ \hat{a}_\uparrow \\ \hat{a}_\downarrow \end{pmatrix} \tag{15}$$

Figure 10 shows, that for $V' \neq 0$ resampling agrees with G-sampling (in this case, gives the correct result), for small $V'$ it gives noisy but correct results, and for $V' = 0$ it gives the wrong result.

## 5 Continuous baths

Systems with an infinite number of non-degenerate bath sites do not produce incomplete estimators of the type discussed in Section 3. However, resampling may still suffer from autocorrelation problems, if a part of the bath is well-approximated by a discrete one, as we will show now.

For this purpose we take an AIM with a bath consisting of two parts: a finite part (which on its own had an incomplete resampling $G$-estimator) and an infinite one, such that for the whole system resampling for $G$ is surjective. Its Hamiltonian is

$$\hat{H} = \hat{H}_{\text{imp}} + \hat{H}_{\text{disc. bath \& hyb.}} + \hat{H}_{\text{cont. bath \& hyb.}} \tag{16}$$

$$= \sum_\sigma (E_\sigma \hat{c}_\sigma^\dagger \hat{c}_\sigma + t \hat{c}_\sigma^\dagger \hat{c}_{\bar{\sigma}})$$

$$+ \sum_\sigma \epsilon_\sigma \hat{a}_\sigma^\dagger \hat{a}_\sigma + \sum_\sigma V(\hat{c}_\sigma^\dagger \hat{a}_\sigma + \hat{a}_\sigma^\dagger \hat{c}_\sigma)$$

$$+ \hat{H}_{\text{cont. bath \& hyb.}} \tag{17}$$

The impurity has again two flavors $\sigma \in \{\uparrow, \downarrow\}$ with respective energy levels $E_\sigma$, connected by a single-particle hopping $t$. Each impurity flavor is connected to a single bath level with energy $\epsilon_\sigma$ and amplitude $V$. For the system $H_{\text{imp}} + H_{\text{disc. bath \& hyb.}}$ alone, critical $Z$-diagrams caused problems with $Z$-sampling, as shown in Sec. 4.1. The Hamiltonian extended with a continuous bath $H_{\text{cont. bath \& hyb.}}$, specifically bands at higher energy, is not supposed to show an incomplete resampling $G$-estimator. Its hybridization function on the real frequency axis is shown in Fig. 11. We transform the hybridization from real frequencies to Matsubara frequencies via

$$\Delta(i\omega_n) = \frac{V^2}{i\omega} + \int_{-6}^{-3} d\epsilon \, \frac{V^2}{i\omega - \epsilon} + \int_{3}^{6} d\epsilon \, \frac{V^2}{i\omega - \epsilon}. \tag{18}$$

For the numerical calculation we set the hybridization strength to $V = 0.3$, $\epsilon_\uparrow = 0.1$, $\epsilon_\downarrow = -0.1$, and especially the hopping $t = 0.2$ to a physically more realistic value.

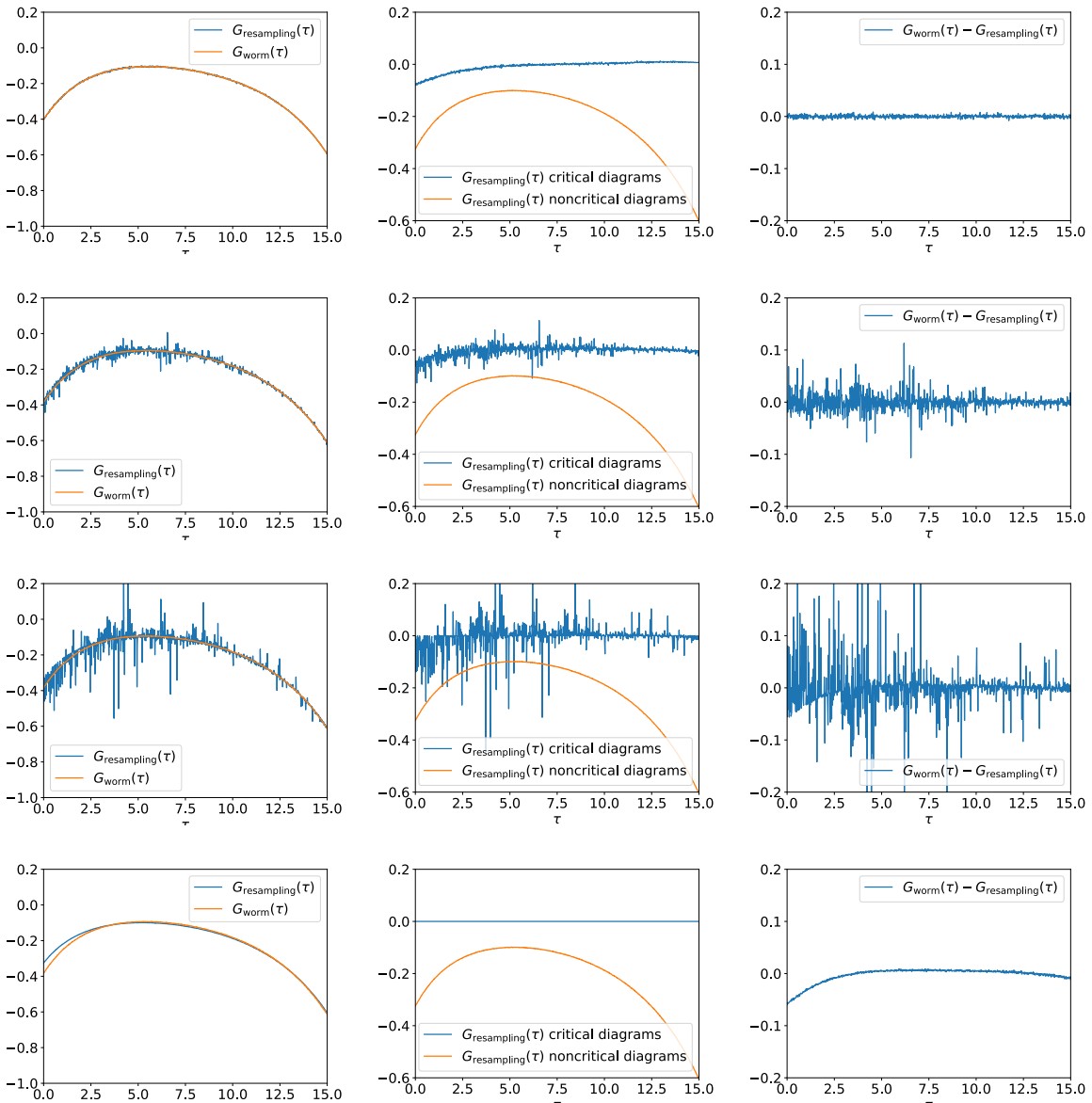

Figure 10: From top to bottom: Hamiltonian $H''[V']$ of Eq. (14) for $V' = 0.03$, 0.003, 0.001, 0.0. The left column shows the Green's functions for resampling and worm sampling, the middle column a distribution of the resampling Green's function in critical and noncritical diagrams, and the right column shows the difference between worm and resampling results.

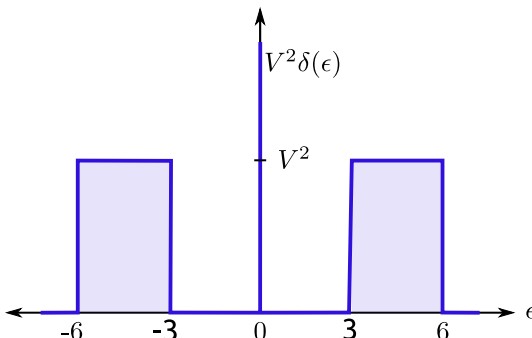

Figure 11: Hybridization function considered for autocorrelation analysis in systems with infinite baths. It comprises of a delta-peak at $\epsilon = 0$ and higher-energy bands. This system is used to mimic a situation with a very narrow peak at the Fermi level and extended hybridization at higher energies.

Let us now investigate the autocorrelation times for this system. As it can be seen in Fig. 12, the worm-sampling produces precise data for all considered inverse temperatures of $\beta = 5$, 10, 15 and 20. We now separate the resampling $G(\tau)$ into two parts, a critical and a non-critical one, with respect to the system $H_{\text{imp}} + H_{\text{disc. bath \& hyb.}}$. For $\beta = 5$ both show the same amount of noise. Upon lowering the temperature, the non-critical part gets smoother, which is to be expected, since the expansion order grows linearly with $\beta$, therefore more information can be extracted from a single diagram. However, the larger expansion order gives the Monte Carlo combinatorically more options to suffer Pauli violations in the bath with respect to the discrete part of $\hat{H}$; this forces the electrons now to propagate through the high energy satellites, and therefore damps the weight of the corresponding $Z$-diagrams significantly. This leads to increased autocorrelation time of the corresponding critical $G$-diagrams, as can be seen in the noise. The results for $Z$-sampling seem to be correct for $\beta = 5$ and 10 and fulfill the sum rule in Eq. (12). The results for $\beta = 15$ and 20 are not converged for $Z$-sampling with this statistics.

This example proves, that the phenomenon described in this work can also significantly affect models with an infinite number of non-degenerate bath sites.

## 6 Conclusion

We found that the standard CTHYB estimator of the Green's function ($Z$-sampling) unexpectedly fails for the Anderson impurity model in some, hitherto unknown, cases. Specifically, this aberration occurs for clusters with $N \geq 2$ flavors on the impurity, where at least one cluster site couples to a discrete, finite bath and this bath does not couple to other impurity flavors through offdiagonal hybridizations. Pauli's exclusion principle forces the weight of some partition function configurations to be zero. This is problematic since nonzero Green's function configurations would need to be generated out of these unreachable partition function configurations. In worm sampling ($G$-sampling), this kind of aberration does not occur.

Furthermore our findings explain the occurrence of large autocorrelation times for systems, whose infinite baths can well be approximated by finite baths, and limit the application of CTHYB for quantum chemistry applications to worm sampling only. In general, our findings illustrate,

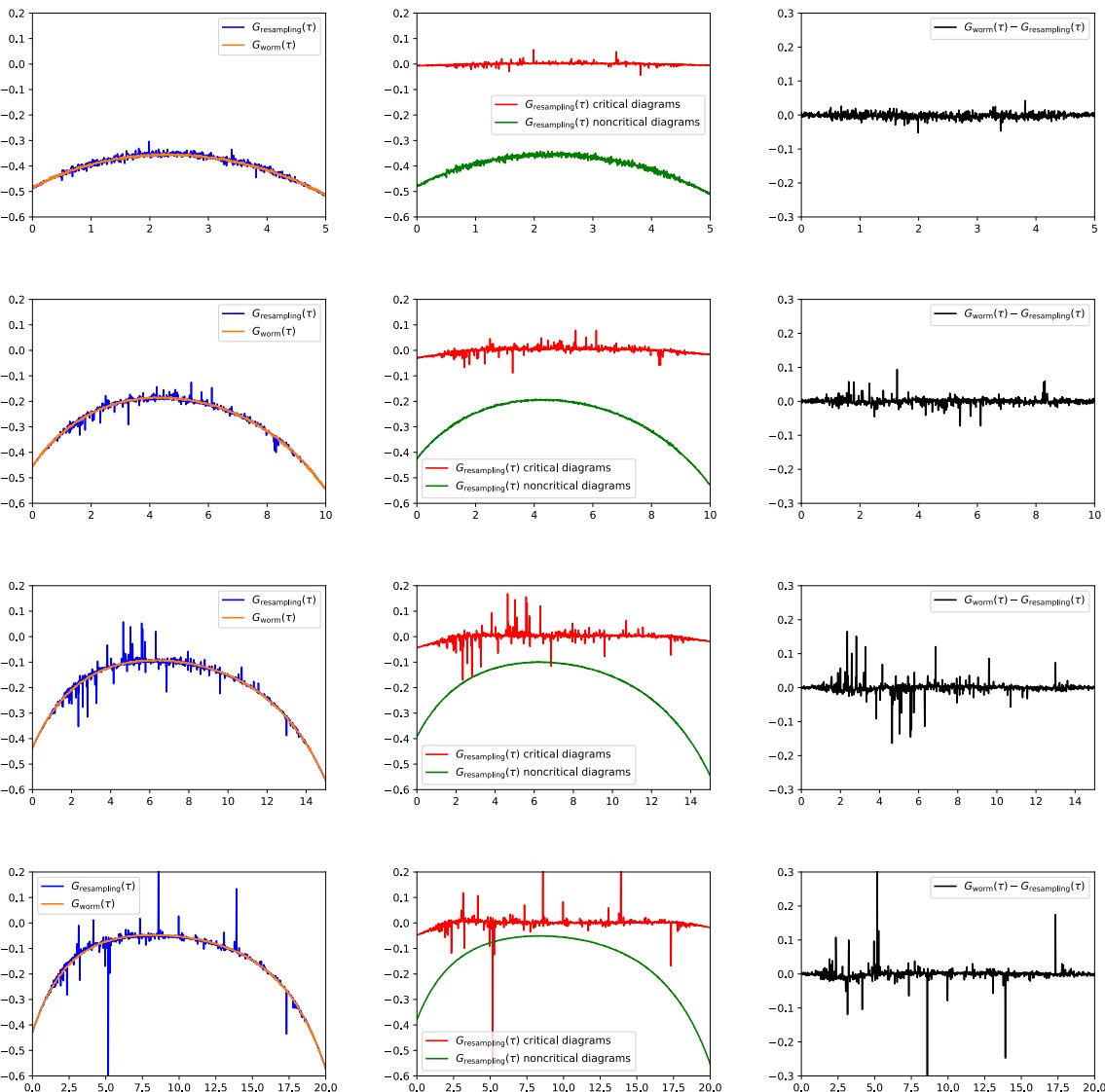

Figure 12: Data for Hamiltonian Eq. 16. From top to bottom: inverse temperatures $\beta = 5, 10, 15$ and $20$. Left column: Greens functions for resampling and worm sampling, middle: splitting of diagrams of resampling in critical and noncritical with respect to the finite subsystem; right: deviation between resampling and worm sampling. It is clear that the noise of the resampling method essentially stems from the critical diagrams.

that for any Markov chain Monte Carlo algorithm of diagrammatic series, it is very important to carefully ensure surjectivity of the mapping between the sampled distribution and observable distribution.

# 7  Acknowledgments

We thank Olivier Parcollet and especially Nils Wentzell for useful discussions. A. H. and G. S. thank the Simons Foundation for the hospitality at the CCQ of the Flatiron Institute, which is a division of the Simons Foundation.

**Funding information**   A. H. and G. S. acknowledge financial support from the DFG through Würzburg-Dresden Cluster of Excellence on Complexity and Topology in Quantum Matter — ct.qmat (EXC 2147, project-id 390858490). M. W. and K. H. acknowledge the FWF (Austrian Science Funds) through project P32044. We further acknowledge funding through the Research Unit "Quast" funded by the DFG as project FOR-5249 (G.S.; project P4) and the FWF as I5868 (K.H.; project P1). We are grateful to the Gauss Centre for Supercomputing e.V. (www.gauss-centre.eu) for funding this project by providing computing time on the GCS Supercomputer SuperMUC-NG at Leibniz Supercomputing Centre (www.lrz.de).

# A  Software

The CTHYB data was produced with w2dynamics [10] using an interface [11] to the TRIQS library [12]. The results shown in Fig. 7 were confirmed with the CTHYB solver from TRIQS [13]. The ED calculations were done with pomerol [14], also using its interface to TRIQS.

# B  Technical remarks

Let us note that the problem of incompleteness discussed here is not caused by an accidental zero of the weight due to numerical noise. However $Z$-configurations with exactly zero weight can have nonzero weight due to numerical instabilities, especially when using Sherman-Morrison formulas for updating the bath weight after insertion / removal of diagram vertices. Then it is possible that the forbidden $G$-configurations can still be accessed with correct probabilities, since the "wrong" but nonzero weights of the $Z$-configurations cancel out. The authors observed such a case, where in resampling $G(\tau)$ first seemingly converged to a wrong result, then it became spiky and showed a very slow convergence towards the correct result. This may happen especially for large expansion orders, i.e. small temperatures. Worm sampling instead immediately converged to the correct result.

## C   Pauli's principle for effective propagators by the example an bath with one site

Here we will discuss how Pauli's principle is implemented for effective propagators using the $Z$-diagram in Fig. 3 (a) as an example. The hybridization function of Hamiltonian (11) is

$$\Delta_{\text{AA}}(\tau) = \frac{V^2}{e^{\beta\epsilon}+1} \times \begin{cases} e^{\epsilon\tau}, & \text{if } \tau > 0 \\ -e^{\epsilon(\beta-\tau)}, & \text{if } \tau < 0 \end{cases}, \tag{19}$$

where the bath site was integrated out. Summing the effective propagators to a determinant for the $Z$-configuration of Fig. 3 (a) gives

$$w_{\text{bath}}(\mathcal{C},\mathcal{C}') = \det \begin{pmatrix} \Delta_{AA}(\tau_1 - \tau_1') & \Delta_{AA}(\tau_1 - \tau_2') \\ \Delta_{AA}(\tau_2 - \tau_1') & \Delta_{AA}(\tau_2 - \tau_2') \end{pmatrix} \tag{20}$$

$$= \left(\frac{V^2}{e^{\beta\epsilon}+1} e^{\epsilon\beta}\right)^2 \det \begin{pmatrix} e^{-\epsilon(\tau_1 - \tau_1')} & e^{-\epsilon(\tau_1 - \tau_2')} \\ e^{-\epsilon(\tau_2 - \tau_1')} & e^{-\epsilon(\tau_2 - \tau_2')} \end{pmatrix}$$

$$= 0, \tag{21}$$

with $\tau_1 < \tau_2 < \tau_1' < \tau_2'$. The matrix in Eq. (21) is rank deficient: the rows are the same, but the first is multiplied with a factor $e^{\epsilon\tau_1'}$, the second with a factor $e^{-\epsilon\tau_2'}$. This is Pauli's exclusion principle in the language of effective propagators. In the actual CTHYB code this is how, at a certain time, only one electron is allowed to propagate through the bath. The generalization to bigger matrices is straightforward.

## D   Energy degenerate bath sites

One may think that the violation of the Pauli principle in a discrete-bath system could be circumvented by adding bath sites that duplicate the existing ones, hence hosting the necessary number of electrons that a given configuration requires. This situation corresponds to the $\delta = 0$ case in Eq. (14). The additional bath sites instead effectively decouple from the impurity as we show in the following by applying a unitary transformation of the bath degrees of freedom.

Let's for simplicity start with the Hamiltonian operator of one impurity and bath site

$$\hat{H} = \begin{pmatrix} \hat{c}^\dagger & \hat{a}_1^\dagger \end{pmatrix} \begin{pmatrix} E & v \\ v & \epsilon \end{pmatrix} \begin{pmatrix} \hat{c} \\ \hat{a}_1 \end{pmatrix} \tag{22}$$

and duplicate the bath site:

$$\hat{H}' = \begin{pmatrix} \hat{c}^\dagger & b_1^\dagger & b_2^\dagger \end{pmatrix} \underbrace{\begin{pmatrix} E & v & v \\ v & \epsilon & 0 \\ v & 0 & \epsilon \end{pmatrix}}_{\mathcal{H}'} \begin{pmatrix} \hat{c} \\ \hat{b}_1 \\ \hat{b}_2 \end{pmatrix}. \tag{23}$$

We can apply the following unitary transformation

$$A = \begin{pmatrix} 1 & 0 & 0 \\ 0 & \frac{1}{\sqrt{2}} & \frac{1}{\sqrt{2}} \\ 0 & \frac{1}{\sqrt{2}} & -\frac{1}{\sqrt{2}} \end{pmatrix}, \tag{24}$$

to the Hamiltonian matrix $\mathcal{H}'$ of $\hat{H}'$ and find

$$A^{\dagger}\mathcal{H}'A = \begin{pmatrix} E & \sqrt{2}v & 0 \\ \sqrt{2}v & \epsilon & 0 \\ 0 & 0 & \epsilon \end{pmatrix}. \tag{25}$$

This object can host two electrons in its bath degrees of freedom, but only one of them is connected to the impurity site with a rescaled hybridization strength of $\sqrt{2}v$. Since Eqs. (23) and (25) are simply the same objects in other bases, in Eq. (23) also only one electron can hop from the impurity into the bath. However, splitting the two bath energies by $\delta$ removes this restriction.

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
