# Peer review of "Aberration of the Green's function estimator in hybridization expansion continuous-time quantum Monte Carlo"

_SciPost Physics_

## Round 1 · Referee Report · Anonymous (Referee 1) · 2023-1-4

Strengths

1 - The work of Andreas Hausoel et al. uncovers a possible deviation of the estimation of the Green's function usually used with hybridization expansion CTQMC. The sampling of the Green's function is vital in order to understand the many body physics of the Anderson impurity model and so to unveil methodological limitations that can occur by using one very common method as CTHYB is very important.

2- The methodology is well introduced and explain in the text.

3 - The numerical results are convincing and the comparison with the worm sampling and exact diagonalization well illustrated.

Weaknesses

1 - The Anderson impurity model is often used in dynamical mean field theory in order to solve the Hubbard model. In this work it is missing a discussion on the implication of this aberration found by the authors. In light of the results presented here I feel that some published work could be put under scrutiny in order to see if their results are correct.

2 - The bibliography is quite limited and I think the authors should add more references to it. There is no reference to the initial work of Anderson ( Anderson, P. W. (1961). "Localized Magnetic States in Metals". Phys. Rev. 124 (1): 41–53) and neither to the first works on the CTHYB (A.N. Rubtsov, A.I. Lichtenstein JETP Lett., 80 (2004), p. 61) but more important a series of articles that can be impacted by these results.

Report

I believe that this work meets the general expectations and criteria to be published. Nevertheless I would recommend to enlarge the implications that this work have on previous works on the Anderson impurity model and on other works on the Dynamical mean field theory.

Requested changes

1 - I would recommend to enlarge the implications that this work have on previous works on the Anderson impurity model and on other works on the Dynamical mean field theory.

2- I would recommend to extend the bibliography especially adding a series of articles that can be impacted by these results.

3-On figure 7 I suggest to add y-axis label for both panels.

4-On figure 9 I suggest to eliminate the x-axis label and ticks label for the first four rows. I also suggest to add y-axis label for all panels.

5-On figure 10 I suggest to eliminate the x-axis label and ticks label for the first three rows. I also suggest to add y-axis label for all panels.

6-On figure 12 I suggest to add the x-axis label on the last row. I also suggest to add y-axis label for all panels. 7- In which way the integrals are evaluated? Probably not but can the results change by using different methods of estimations of the integrals?

---

## Round 1 · Referee Report · Anonymous (Referee 2) · 2023-1-18

Strengths

1- The paper “Aberration of the Green's function estimator in hybridization expansion continuous-time quantum Monte Carlo” by Andreas Hausoel, Markus Wallerberger, Josef Kaufmann, Karsten Held, Giorgio Sangiovanni presents specify the problem in the standard CTHYB estimator of the Green's function, and identify the origin, which is related to the Pauli principle constraints in the Anderson impurity model. They suggest an alternative Green's function estimator, which works well in comparison with the exact diagonalization. 2- The underlying idea for the method is well described. 3- Numerical results support their claim.

Weaknesses

1- There was no indication of how V, δ, etc. should be set quantitatively, which might be depend on the other parameters. 2-No discussion existed on how this method affects the self-consistency calculations when CTQMC is used as an impurity problem for DMFT. 3-The specific input to reproduce the numerical results given in the manuscript is not on the Github. It would be very helpful for readers if the authors provide the ready-to-run input. Since there seems only DMFT examples exists currently.

Report

Overall, this is a well-written paper that makes an important contribution to the field. I recommend that it be accepted for publication in SciPost physics with the following major revisions.

Requested changes

[Major] 1- As shown in the Figures 9-11, the behavior of the critical diagrams contribution to G(\tau) is quite different for the parameters such as V, \delta and \beta. It would be of interest, how the validity for these parameters changes in the AIM. 2- I understand that it is an improvement in the Green's function obtained by CTHYB in the high-temperature region, but it is a hindrance to systematic analysis on the high energy region to the low energy region, which is the strength of CTHYB, and it seems one need to turn on and off this estimator as approaching to the low-energy region. Do the authors have any solutions regarding this point? At least a comment on this point is needed. 3- Also, it would be very helpful for the readers to give the affect of the self-consistency calculations when this CTQMC is used as an impurity problem for DMFT.

[Minor] 3- On page 18, authors refer to Fig. 3 (a), which is not exists.

---

## Editorial Decision

awaiting_resubmission